# Long Intergenic Non-Coding RNAs and *BRCA1* in Breast Cancer Pathogenesis: Neighboring Companions or Nemeses?

**DOI:** 10.3390/ncrna11010009

**Published:** 2025-01-29

**Authors:** Olalekan Olatunde Fadebi, Thabiso Victor Miya, Richard Khanyile, Zodwa Dlamini, Rahaba Marima

**Affiliations:** 1SAMRC Precision Oncology Research Unit (PORU), DSI/NRF SARChI Chair in Precision Oncology and Cancer Prevention (POCP), Pan African Cancer Research Institute (PACRI), University of Pretoria, Hatfield 0028, South Africa; 2Department of Medical Oncology, Faculty of Health Sciences, Steve Biko Academic Hospital, University of Pretoria, Hatfield 0028, South Africa

**Keywords:** breast cancer, *BRCA1*, lincRNAs, diagnosis, biomarker, signaling pathways

## Abstract

Breast cancer is one of the leading causes of mortality among women, primarily due to its complex molecular landscape and heterogeneous nature. The tendency of breast cancer patients to develop metastases poses significant challenges in clinical management. Notably, mutations in the breast cancer gene 1 (*BRCA1*) significantly elevate breast cancer risk. The current research endeavors employ diverse molecular approaches, including RNA, DNA, and protein studies, to explore avenues for the early diagnosis and treatment of breast cancer. Recent attention has shifted towards long non-coding RNAs (lncRNAs) as promising diagnostic, prognostic, and therapeutic targets in the multifaceted progression of breast cancer. Among these, long intergenic non-coding RNAs (lincRNAs), a specific class of lncRNAs, play critical roles in regulating various aspects of tumorigenesis, including cell proliferation, apoptosis, epigenetic modulation, tumor invasion, and metastasis. Their distinctive expression patterns in cellular and tissue contexts underscore their importance in breast cancer development and progression. Harnessing lincRNAs’ sensitivity and precision as diagnostic, therapeutic, and prognostic markers holds significant promise for the clinical management of breast cancer. However, the potential of lincRNAs remains relatively underexplored, particularly in the context of *BRCA1*-mutated breast cancer and other clinicopathological parameters such as receptor status and patient survival. Consequently, there is an urgent need for comprehensive investigations into novel diagnostic and prognostic breast cancer biomarkers. This review examines the roles of lincRNAs associated with *BRCA1* in the landscape of breast cancer, highlighting the potential avenues for future research and clinical applications.

## 1. Introduction

Breast cancer accounts for 11.6% of cancer-related deaths among women worldwide (Figure 1) [1]. It is the most frequently diagnosed malignancy in women globally [2,3], and the primary factor contributing to low survival rates is distant metastasis [4]. Breast cancer is a complex disease influenced by genetic, epigenetic, environmental, and lifestyle factors. While most breast cancer cases are sporadic, approximately 10% are attributable to genetic mutations, including those in the breast cancer gene 1 (*BRCA1*) and breast cancer gene 2 (*BRCA2*).

Family history plays a critical role in breast cancer risk, as germline mutations in the tumor suppressor genes *BRCA1* and *BRCA2* are strongly associated with an inherited predisposition to the disease [5]. Incidence rates continue to rise, particularly in developed nations, where factors such as aging, hormonal and chemical exposures, and systemic disorders contribute to elevated risk [5]. Women are significantly more likely than men to develop breast cancer.

Extensive research has been conducted on the molecular mechanisms by which tumor cells acquire metastatic traits, leading to significant advancements in diagnostic and prognostic techniques. Despite these efforts, the persistently high mortality rate associated with breast cancer remains a substantial challenge [6,7,8].

Breast cancer gene expression profiles are classified into three main molecular subtypes: basal, human epithelial growth factor receptor 2 (HER2)-enriched, and luminal A/B [9]. Clinical practice utilizes the status of estrogen receptor (ER), progesterone receptor (PR), and HER2 hormone receptors to categorize breast cancer, enabling the optimization of personalized therapeutic strategies for each patient [9,10].

The HER2-enriched subtype accounts for 15–20% of all breast cancer cases, while the luminal subtype constitutes approximately 75%. Luminal breast cancer is characterized by ER expression and the absence of HER2 growth factor expression [11,12]. In contrast, the HER2-enriched subtype demonstrates more aggressive tumor behavior and accelerated growth, as it is negative for hormone receptors but positive for HER2 expression [13]. Triple-negative and basal-like breast cancer are subtypes that do not express ER, PR, or HER2 receptors; while these categories are distinct, they are often used interchangeably in the literature [14].

The human genome contains approximately 20,000 protein-coding genes, representing less than 2% of the genome [15]. Among the identified classes of non-coding RNA (ncRNA) are long intergenic non-coding RNAs (lincRNAs), which range in size from a few hundred to tens of thousands of bases (≥200). Despite identifying over 3000 human lincRNAs, only about 1% have been fully characterized [11].

While lincRNAs have been implicated in several cancer types, including *BRCA1*-mutated breast cancer, their therapeutic significance in breast cancer remains unclear. Understanding the roles of lincRNAs in *BRCA1*-driven metastasis is critical to improving breast cancer therapies. This review discusses the mechanisms involving lincRNAs and *BRCA1* in breast cancer pathogenesis, highlighting their potential for advancing treatment strategies.

## 2. *BRCA1* Physiological Role and Regulation

*BRCA1* is a commonly occurring tumor suppressor gene that encodes a nuclear protein with a molecular weight of 220 kDa. It is found in at least 5% of unselected patients with breast cancer [16]. *BRCA1* plays a vital role in cell cycle control, replication fork protection, DNA repair, and the regulation of gene transcription (Figure 2a).

A significant increase in the incidence of breast cancer and ovarian cancer is observed when *BRCA1* is deleted or mutated. This gene is essential for maintaining genomic stability, a critical factor in preventing carcinogenesis. As a multifunctional gene, *BRCA1* is integral to the cellular DNA damage response and repair pathways, underscoring its importance in tumor suppression.

*BRCA2* is a related gene to *BRCA1* [17]. Both act as cancer-predisposing genes discovered through the genetic studies of familial breast cancer [18]. Cells lacking functional *BRCA1*/*BRCA2* proteins show a significantly compromised ability to repair DNA double-strand breaks (DSBs) via homologous recombination (HR) [19]. As a result, cancers with mutations in *BRCA1/2* are highly susceptible to treatments that damage DNA, such as poly (ADP-ribose) polymerase (PARP) inhibitors [19,20]. The tumor phenotype varies in breast cancer patients based on *BRCA1* or *BRCA2* germline mutations. Unlike *BRCA2* mutation carriers, who are more likely to develop tumors that are progesterone receptor (PR)- or estrogen receptor (ER)-positive, *BRCA1* mutation carriers predominantly develop triple-negative breast cancer (TNBC) (Figure 2b) [21].

*BRCA1* consists of multiple functional domains that interact with other tumor suppressor genes involved in various DNA repair pathways, including HR, single-strand annealing (SSA), and non-homologous end joining (NHEJ) [17]. *BRCA1* plays a critical role in mediating the recruitment of *RAD51* proteins to the sites of DNA damage during DSBs. This recruitment is facilitated by *PALB2*, which connects *BRCA1* and *BRCA2*, ultimately leading to HR-mediated DSB repair [18]. While *BRCA1* has been extensively studied for its roles in stimulating and suppressing repair mechanisms, its precise function in NHEJ remains to be fully understood [19,20].

Additionally, *BRCA1* plays a significant role in G1/S and G2/M checkpoint activation. Cells must remain in cycle arrest when DNA damage occurs to facilitate repair. If this process fails, genetic mutations can accumulate, leading to chromosomal abnormalities and the development of malignant cells [21,22]. Carriers of a single germline mutation in *BRCA1* face a heightened risk of developing various cancers, including breast cancer [23]. Individuals with *BRCA1* germline mutations are more susceptible to breast cancer, with up to 35% of newly diagnosed breast cancer patients harboring *BRCA1* mutations. These individuals are three times more likely to experience early-onset breast cancer [24]. Tumors without *BRCA1* or *BRCA2* germline mutations are called “BRCAness”. These tumors display characteristics similar to *BRCA*-mutated tumors due to a deficit in HR [25,26]. Intracellular and intercellular signaling pathways in breast cancer are influenced by non-coding RNAs (ncRNAs) [27]. Long non-coding RNAs (lncRNAs) are further categorized into various subtypes, providing insights into their underlying mechanisms of action and functional significance.

## 3. *BRCA1*-Mutated Breast Cancer

*BRCA1*-mutated breast cancer represents a significant subset of cases, primarily defined by its genetic basis and associated clinical outcomes. Although *BRCA1* mutations account for only 2–3% of all breast cancer diagnoses in the general population, they contribute to 40–45% of the hereditary breast cancer cases [28]. Female carriers of *BRCA1* mutations have a 70% lifetime risk of developing breast cancer by the age of 80 [29]. *BRCA1*-mutated breast cancer predominantly belongs to the basal-like subtype, as identified through gene expression profiling. High-grade histopathological features are frequently observed, including a high mitotic index, pushing tumor edges, and pleomorphic nuclei [30]. *BRCA1* mutations are most commonly associated with triple-negative breast cancer (TNBC), a subtype lacking the expression of estrogen receptor (ER), progesterone receptor (PR), and human epidermal growth factor receptor 2 (HER2). Aggressive behavior, high-grade tumors, and poor prognosis characterize TNBC. While TNBC accounts for 15–25% of all breast cancer cases, this percentage is significantly higher among *BRCA1* mutation carriers [31]. In breast cancer, *BRCA1* mutations disrupt homologous recombination (HR) repair processes, leading to increased genomic instability and the accumulation of DNA double-strand breaks (DSBs). This instability drives the aggressive nature and carcinogenesis of *BRCA1*-mutated breast cancer [19]. Although much attention has been given to protein-coding gene mutations, non-coding RNAs (ncRNAs) are increasingly recognized as critical in understanding cancer biology. The interplay between *BRCA1*-mutated breast cancer and ncRNAs has recently garnered significant interest. ncRNAs, including microRNAs (miRNAs), long non-coding RNAs (lncRNAs), and circular RNAs (circRNAs), regulate gene expression, often at post-transcriptional or epigenetic levels [15]. The dysregulation of these ncRNAs has been implicated in key cancer characteristics such as invasion, proliferation, and treatment resistance, particularly in the context of *BRCA1* mutations [17]. Its HR-deficient (HRD) nature heavily influences treatment strategies for *BRCA1*-mutated breast cancer. Targeted therapies, such as poly (ADP-ribose) polymerase (PARP) inhibitors, exploit this vulnerability. Drugs like olaparib have shown promise in improving outcomes for individuals with hereditary *BRCA1*-related cancers, including breast cancer and ovarian cancer.

## 4. LincRNAs Association with *BRCA1* in Breast Cancer

Ding et al. conducted RNA sequencing on breast cancer tissue samples and the corresponding adjacent normal tissues to investigate whether the aberrant expression of long intergenic non-coding RNAs (lincRNAs) is associated with cancer [32]. Among the lincRNAs analyzed, most were expressed in the tissues adjacent to cancer, while a smaller subset was expressed in cancerous tissues. Notably, some lincRNAs displayed lower expression levels in breast cancer tissues than the adjacent normal tissues, while others exhibited higher expression in healthy tissues. The study revealed that lincRNA-BC5 expression was positively correlated with PR concentration, clinical stage, and patient age, whereas lincRNA-BC8 expression was negatively correlated with PR expression. Furthermore, advanced breast cancer grade was associated with an increased expression of *lincRNA-BC4* [32]. The advent of advanced high-throughput deep sequencing technologies has enabled the comprehensive profiling of lincRNAs. These technologies allow identifying unique, tissue-specific lincRNAs and provide quantitative data, even for low-abundance species, as the frequency of sequencing reads reflects their relative abundance. Using deep sequencing, Ding et al. identified lincRNA expression profiles in five pairs of snap-frozen breast cancer tissues and their matched adjacent normal tissues. The analysis revealed significant alterations in lincRNA expression between the cancerous and adjacent tissues, suggesting that aberrantly expressed lincRNAs may play roles in carcinogenesis [32]. Specifically, *lincRNA-BC2* and *lincRNA-BC5* showed increased expression in breast cancer tissues compared to matched normal tissues, whereas *lincRNA-BC4* and *lincRNA-BC8* were downregulated. These findings were consistent with the deep sequencing results [32]. However, other lincRNAs identified, such as *lincRNA-BC7*, remain uncharacterized concerning their role in breast cancer and *BRCA1*.

## 5. The Molecular Mechanisms of lincRNAs and *BRCA1* in Breast Cancer Metastasis

Over the past decade, extensive research has sought to unravel how lincRNAs mediate cancer invasion and metastasis. Evidence suggests that lincRNAs may regulate metastatic genes and their activity at various levels, from transcriptional to translational [33]. Additionally, lincRNAs influence the spread of breast cancer by modulating multiple cell signaling pathways. The current understanding of the functional mechanisms by which lincRNAs are involved in the invasion and metastasis of breast cancer cells is outlined in the following subsections. Functionally, lincRNAs can act as either the facilitators or suppressors of breast cancer cell invasion and metastasis, highlighting their dual roles in cancer progression.

### 5.1. LincRNAs Facilitate the Invasion and Metastasis of Breast Cancer Cells

Certain types of breast cancer cells exhibit the aberrant upregulation of specific lincRNAs [33,34,35,36]. Numerous studies have identified upregulated lincRNAs, including *BREAST CANCERR4*, *linc-ROR*, and HOX transcript antisense RNA (*HOTAIR*), as the key breast cancer invasion and metastasis facilitators.

*HOTAIR* was the first lincRNA discovered with the ability to accelerate tumor growth. Early investigations revealed that *HOTAIR* promotes breast cancer metastasis and is highly expressed in metastatic breast cancer tissues [37]. Mechanistically, *HOTAIR* recruits the PRC2 complex to target genes across the genome, altering histone H3 methylation at lysine 27. This epigenetic modification silences metastasis-suppressor genes, thereby increasing cancer invasiveness and spread [37]. The interaction between *HOTAIR* and PRC2 has significant therapeutic implications for breast cancer metastasis. However, the precise biochemical pathways through which *HOTAIR* regulates PRC2 remain unclear. Collina et al. found a significant correlation between *HOTAIR* overexpression and lymph node metastases in TNBCs [38]. Despite these findings, further research is needed to clarify HOTAIR’s roles in the initiation, progression, and metastasis of breast cancer. Surprisingly, *linc-ROR* also uniquely regulates breast cancer metastasis and epithelial–mesenchymal transitions (EMTs) [39]. A study by Mondal and Meeran demonstrated that *linc-ROR* knockdown in breast cancer cells inhibited breast cancer-lung metastasis in vivo in immunodeficient mice. Conversely, *linc-ROR* overexpression enhanced the EMT, migration, and invasion of breast cancer cells [40]. The study also revealed that *linc-ROR* acts as a competing endogenous RNA (ceRNA), modulating miR-205 activity to prevent the degradation of its target genes, which otherwise suppress breast cancer lung metastasis [40]. Fan et al. further identified that *linc-ROR* functions as a decoy lincRNA, preventing the recruitment of G9A methyltransferase and chromatin regulatory factors. This mechanism eliminates the histone H3K9 modification of the TESC (tescalcin) promoter, facilitating breast cancer spread [41]. Despite these insights, additional studies are needed to confirm the genome-wide *linc-ROR* occupancy profile and investigate other epigenetic markers targeted by *linc-ROR* in breast cancer metastasis. Moreover, *Linc-ROR* directly interacts with heterogeneous nuclear ribonucleoprotein I (hnRNP I), serving as a potent negative regulator of p53. This interaction inhibits p53-mediated cell cycle arrest and apoptosis in breast cancer cells, further contributing to disease progression (Figure 3) [42].

The evidence suggests that the RoR–hnRNP I–p53 axis may play a critical role in breast cancer metastasis. Among patients with advanced breast cancer, lincRNA *BREAST CANCERR4* is highly expressed and significantly contributes to metastasis. *BREAST CANCERR4* promotes the spread of breast cancer by binding to two transcription factors, *SNIP1* and *PNUTS*, in response to chemokines. This interaction exerts long-lasting regulatory effects by activating a noncanonical hedgehog/GLI2 transcriptional program, which drives cell migration [43].

Liu et al. speculated that lincRNA *BREAST CANCERR4* might also be vital in regulating metastasis in other cancers via the hedgehog signaling pathways, as these pathways are aberrantly active across multiple cancer types [43]. Additionally, a growing body of research has identified numerous lincRNAs that facilitate breast cancer metastasis, including *lnc-SLC4A1-1* [44], *Lnc-BM* [45], *BLACAT1* [46], *H19* [47,48], and terminal differentiation-induced ncRNA (*TINCR*) [49], among others [50,51]. These findings prove lincRNAs’ essential role in enhancing breast cancer cells’ invasive and metastatic potential.

### 5.2. LincRNAs Function to Inhibit the Invasion and Spread of Breast Cancer Cells

Research has identified several lincRNAs, such as *MALAT1*, *NKILA*, and *ANCR*, that inhibit breast cancer metastasis, in contrast to lincRNAs known to promote it [43]. One of the most conserved lincRNAs, *MALAT1*, is widely expressed in healthy tissues [52,53]. Initially recognized as a biomarker for early-stage non-small cell lung cancer capable of predicting metastasis and survival [54], *MALAT1* exhibits a complex role in breast cancer progression. Its deletion has been linked to increased metastatic potential in breast cancer, whereas overexpression in transgenic, xenograft, and syngeneic models has been shown to reduce metastasis [55]. Conversely, knockdown studies using *MALAT1* antisense oligonucleotide (ASO) in breast cancer cell lines and animal models decreased metastasis and impaired cell migration [56]. These seemingly contradictory findings highlight the multifaceted nature of *MALAT1*’*s* function in breast cancer metastasis.

LncRNA *NKILA* has similarly been associated with reduced breast cancer metastasis despite its correlation with poor patient outcomes [57]. Mechanistically, *NKILA* stabilizes the NF-κB/IκB complex by binding directly to it and masking IκB’s phosphorylation sites, thereby inhibiting NF-κB signaling [43]. This direct interaction emphasizes *NKILA*’*s* unique role in regulating signal transduction without recruiting additional regulatory proteins. lncRNA *ANCR* has also emerged as a repressive regulator of breast cancer metastasis. Studies in immunodeficient mice demonstrated *ANCR’s* ability to suppress metastasis in vivo [58]. Mechanistically, *ANCR* interacts with EZH2, facilitating CDK1 binding, phosphorylation, and the subsequent EZH2 degradation [58]. Additionally, *ANCR* regulates the TGF-β pathway, contributing to reduced lung metastasis from breast cancer. Interestingly, TGF-β1 treatment did not alter EZH2 mRNA or protein levels, suggesting that ANCR may modulate TGF-β1-induced metastasis through mechanisms beyond EZH2 degradation [58,59]. Further research has uncovered additional tumor-suppressor lincRNAs, including *PDCD4-AS1* [59], *LINC01133* [60], *NORAD* [61], *MEG3* [62], and *XIST* [63,64,65]. These findings suggest that lincRNAs hold significant potential as biomarkers and therapeutic targets for mitigating breast cancer metastasis.

### 5.3. LincRNAs as Regulators of BRCA1

Recent research highlights the critical role of lincRNAs in epigenetic, transcriptional, and post-transcriptional gene regulation [66,67]. LincRNAs are tightly regulated and form intricate secondary structures, enabling them to interact with proteins and nucleic acids. These interactions allow lincRNAs to influence various biological processes by modulating protein–nucleic acid dynamics. Notably, lincRNAs have been shown to regulate *BRCA1* through several mechanisms, including DNA double-strand repair [32], IR-induced apoptosis, cell cycle regulation, and the modulation of the Wnt/β-catenin pathway [68,69,70].

Emerging evidence also underscores the association between lincRNAs and breast cancer. Ding et al. identified 538 lincRNAs with differential expression in breast cancer tissues; however, the differences in expression between breast cancer subtypes were not disclosed [32]. The dysregulated expression of *HOTAIR* in *BRCA1* has been observed in breast cancer (Table 1) [71,72]. Additionally, Merry et al. identified three dysregulated lincRNAs due to the amplification of receptor tyrosine-protein kinase ERBB2, a driver of increased tumorigenic potential in breast cancer [73]. Table 1 provides a comprehensive overview of lincRNAs implicated in breast cancer, detailing their roles and expression patterns. This growing body of evidence underscores the potential of lincRNAs as crucial regulators and biomarkers in breast cancer pathogenesis, paving the way for further exploration into their therapeutic applications.

LincRNAs play a significant role in regulating EMT, a process that accelerates the development and metastasis of breast cancer [39]. Despite the complexity and heterogeneity of *BRCA1*-mutated breast cancer, only 18 lincRNAs associated with breast cancer have been annotated in the lincRNA Disease database for lincRNA-related disorders [78]. Among these, the lincRNA neighbor of *BRCA1 gene 2* (*NBR2*), located adjacent to the *BRCA1* gene, has garnered particular attention [74]. Initially, *NBR2* was thought to be co-mutated or deleted alongside *BRCA1* in malignancies due to the high mutation/deletion rate of *BRCA1* in breast and ovarian cancers. Subsequent studies confirmed *NBR2*’*s* identity as a lincRNA and have begun to elucidate its intricate roles in tumor biology, revealing its involvement in tumor suppression alongside BRCA1 [79].

In regulating *BRCA1*, antisense ncRNA in the INK4 locus (*ANRIL*) is one of the most extensively studied lincRNAs. *ANRIL* recruits PRC2 to the *BRCA1* promoter, epigenetically silencing *BRCA1* expression. This downregulation contributes to genomic instability and breast cancer progression, as it compromises HR repair [74]. Similarly, the well-characterized lincRNA *HOTAIR* indirectly influences *BRCA1* by altering chromatin states. *HOTAIR* facilitates the recruitment of restrictive histone modifiers, such as EZH2, to *BRCA1* regulatory regions, suppressing its transcription. High *HOTAIR* expression is associated with poor prognosis and resistance to DNA-damaging therapies in breast cancer [74]. Another key lincRNA, *linc-ROR*, modulates the EMT pathway by interacting with *BRCA1*. *Linc-ROR* inhibits *BRCA1* expression, promoting a mesenchymal phenotype and increasing breast cancer metastasis risk. Its role in treatment resistance underscores its therapeutic relevance [74]. Additionally, taurine-upregulated gene 1 (*TUG1*) interacts with chromatin-modifying complexes to suppress *BRCA1*. The overexpression of *TUG1* disrupts *BRCA1*-mediated DNA repair pathways, contributing to genomic instability. Targeting *TUG1* represents a potential therapeutic strategy to restore BRCA1 activity in breast cancer [74]. These examples illustrate how lincRNAs regulate *BRCA1* through diverse mechanisms, influencing its expression, stability, and function. By modulating *BRCA1*, lincRNAs play crucial roles in maintaining genomic integrity and tumor suppression, highlighting their potential as therapeutic targets in breast cancer.

## 6. The Role of lincRNAs as a Biomarker in Breast Cancer Diagnosis

Despite advancements in diagnostic techniques, a reliable strategy for diagnosing cancer patients remains elusive. Therefore, the development of innovative diagnostic methods that facilitate early detection and minimize overdiagnosis and over-treatment is urgently needed. One of the most promising non-invasive techniques for early detection is the screening of biomarkers in extracellular fluids. LincRNAs, being highly stable in bodily fluids—particularly when incorporated into exosomes or apoptotic bodies—are ideal candidates for cancer diagnostic and prognostic biomarkers [80]. Research has shown that despite ribonucleases in bodily fluids, lincRNAs capable of withstanding ribonuclease degradation have been identified in samples such as serum, cerebrospinal fluid, and urine [80]. Furthermore, the dysregulation of lncRNA observed in primary tumor tissues is mirrored in various body fluids, including plasma, urine, saliva, whole blood, and gastric juice [81]. Investigations into the expression levels of lincRNAs in breast cancer tissues compared to normal tissues suggest that several of these could serve as potential diagnostic biomarkers. For example, while *lincRNA-BC4* and *lincRNA-BC8* were consistently downregulated, *lincRNA-BC2* and *lincRNA-BC5* were upregulated (more than twice) in breast cancer samples [32]. Additionally, *lincRNA-BC4* expression was notably lower in grade III breast cancer, whereas *lincRNA-BC5* expression was significantly higher in grade III breast cancer. Moreover, *lincRNA-BC2* expression was strongly correlated with lymph node metastasis (LNM). Research has also demonstrated that lincRNAs are detectable in physiological fluids. For instance, breast cancer patients showed significantly higher serum expression levels of the circulating lncRNA *RP11-445H22.4*, allowing breast cancer detection with 74% specificity and 92% sensitivity [82,83]. Bermejo et al. found that peripheral blood from TNBC patients exhibited hypermethylated lincRNA *LINC00299*. Additionally, breast cancer patients had significantly elevated blood levels of lincRNA *LINC00310*. The examination of the receiver operating characteristic (ROC) curve revealed that lincRNA *LINC00310* effectively distinguished between breast cancer patients and healthy individuals (area under curve = 0.828) [84]. Another study comparing plasma from breast cancer patients to healthy controls found elevated levels of lincRNAs *H19*, *HOTAIR*, and *RP11-445H22.4* [85].

## 7. LincRNAs as Breast Cancer Prognostic Markers

Recent research suggests that lincRNAs may serve as valuable prognostic indicators, particularly for breast cancer patients with *BRCA1* mutations. There is a strong correlation between the expression patterns of different lincRNAs, patient prognosis, and the progression of breast cancer. For instance, in breast cancer tissues, elevated levels of lincRNA *H19* are linked to more advanced disease stages and poorer prognoses [74,75]. This suggests that *H19* may act as a predictive biomarker for aggressive breast cancer morphologies. Additionally, other lincRNAs, such as *LINC00657* and *LINC01087*, have been associated with low overall survival rates in breast cancer patients, with overexpression linked to poor prognosis [74,75]. *LINC00346* is another lincRNA associated with poor prognosis, potentially reflecting *BRCA1* status [74]. Furthermore, *LINC00894* promotes cancer cell invasion and proliferation, with high expression correlating with a worse overall survival rate due to its impact on *BRCA1* pathways [74]. *LINC00574* serves as a prognostic marker, with its overexpression linked to recurrence in breast cancer patients [74]. This suggests its involvement in *BRCA1*-related pathways, contributing to tumor aggressiveness. *LINC00323* has also been implicated in controlling cell growth and death in breast cancer, with its expression serving as a potential indicator of *BRCA1*’*s* functional state and patient prognosis [32]. These findings underscore the complex interplay between lincRNAs and *BRCA1* function in the pathophysiology of breast cancer. Disruptions in lincRNA expression can impact DNA repair processes and carcinogenesis, increasing breast cancer risk and progression. Thus, lincRNAs represent critical, independent prognostic indicators and modulators of *BRCA1*-related pathways. Exploring their roles offers promising insights into breast cancer biology and potential therapeutic targets.

## 8. Limitations of Using LincRNAs in *BRCA1* Breast Cancer Pathogenesis

The use of lincRNAs in the pathophysiology of *BRCA1*-associated breast cancer remains limited, primarily due to the lack of functional validations in the existing lincRNA databases. This impedes researchers from further conclusions about aberrant lincRNA expression’s role in carcinogenesis. Additionally, the inability to replicate subtype-specific lincRNAs is challenging, stemming from limited sample sizes and insufficient subtype data during replication. Moreover, a comprehensive worldwide comparison of gene expression profiles across breast tumors, adjacent tissues, and normal tissues is inadequate due to the heterogeneity of tumors and cell mixing within tumor tissues. This is particularly true for lincRNAs, whose expression patterns exhibit broad tissue- or cell-type specificity. Consequently, transcriptome analysis focused on single cells or sorted cell populations is crucial for accurately identifying these lincRNAs as reliable biomarkers [86].

## 9. Conclusions and Future Perspectives

This work highlights the dual role of lincRNAs about *BRCA1* as both potential allies and adversaries in breast cancer development. On one hand, certain lincRNAs exhibit oncogenic properties, contributing to tumor progression and poor patient outcomes. On the other hand, others may act as tumor suppressors, reflecting the complexity of their functions within the tumor microenvironment. Understanding the intricate interactions between the *BRCA1* gene and lincRNAs is crucial for elucidating the pathophysiology of breast cancer. Specifically, lincRNAs such as *NBR2* play dual roles in breast cancer pathogenesis—acting either as suppressors by modulating BRCA1 expression and activity or as promoters of breast cancer development through the dysregulation of essential pathways. This complex relationship underscores the intricacies of gene regulation in cancer biology and the urgent need to understand these processes better.

LincRNAs play a significant role in the development of breast cancer and its response to the subsequent treatment, making them a promising therapeutic target with substantial potential for effectiveness in breast cancer theranostics. Despite the challenges associated with in vivo research, lincRNAs remain valuable as prospective therapeutic targets and indicators. However, their minimal sequence conservation may limit their translational applications. Future advancements in breast cancer treatment and biomarker discovery are contingent on further exploration of lincRNAs about *BRCA1*. Understanding the specific regulatory networks involving lincRNAs could pave the way for innovative strategies to restore normal gene expression patterns or inhibit oncogenic pathways. Further research is essential for the functional characterization of lincRNAs and their interactions within the tumor microenvironment, especially under varying physiological conditions such as hypoxia or exposure to chemotherapeutic agents. Additionally, more studies are required to evaluate the application of lincRNAs as diagnostic and prognostic biomarkers and therapeutic targets in breast cancer to address the current limitations in diagnosis and therapy approaches.

## Figures and Tables

**Figure 1 ncrna-11-00009-f001:**
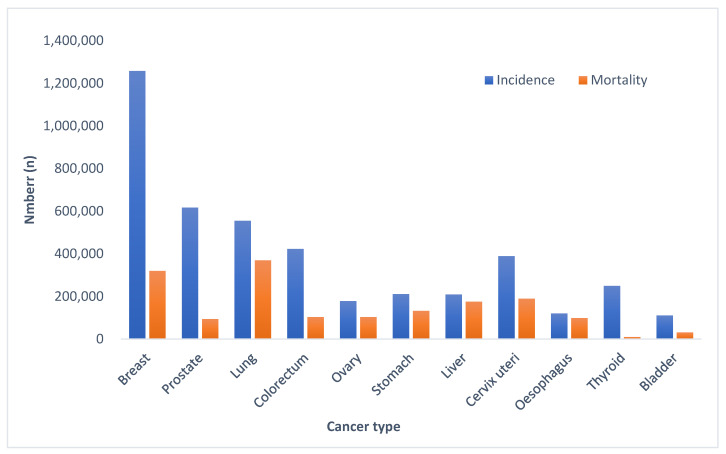
Estimated cancer incidence and mortality rates worldwide in 2022. Data Source: GLOBOCAN 2022, Graph Production: Global Cancer Observatory http://gco.iarc.fr (accessed on 8 October 2024). International Agency for Research on Cancer.

**Figure 2 ncrna-11-00009-f002:**
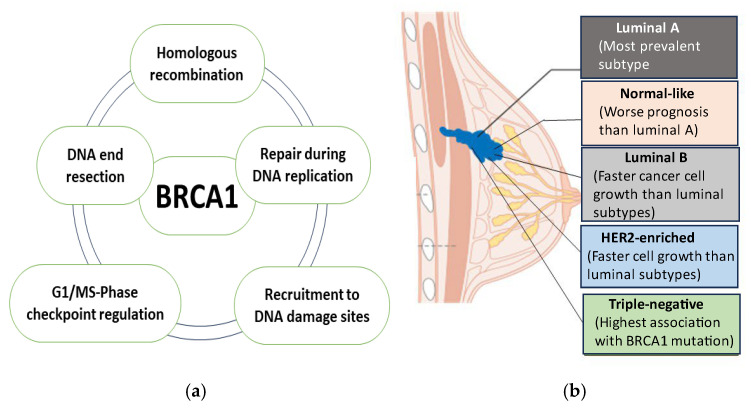
(**a**) Main roles of *BRCA1*. An overview of *BRCA1*’*s* physiological functions in HR, repair during the DNA-replication recruitment of DNA damage site, G1/MS-phase checkpoint regulation, and DNA end resection. (**b**) *BRCA1*-mutated breast cancer associated with TNBC based on their expression of hormone receptors (ER, PR, and HER2). The image was self-created with BioRender.com.

**Figure 3 ncrna-11-00009-f003:**
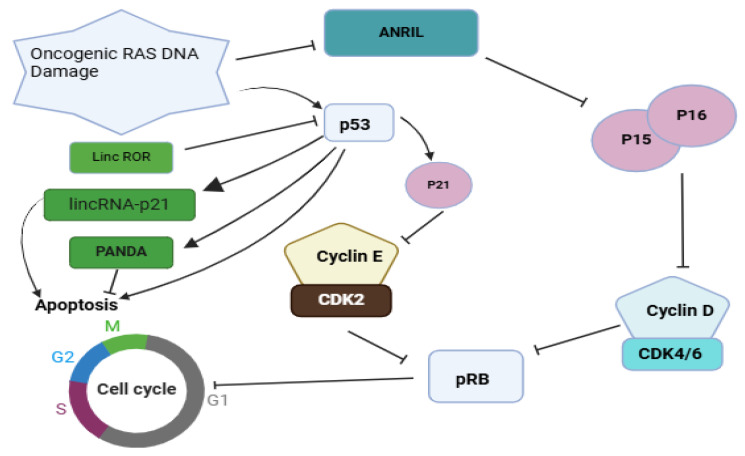
Regulation of the pRB and p53 pathways by the lincRNAs *ANRIL*, *lincRNA-p21*, and *lincRNA-ROR*. These lincRNAs regulate the pRB and p53 pathways by modulating the expression of CDK inhibitors p15 and p16, which impact pRB activity. *ANRIL* acts as a coregulator by binding to polycomb proteins. In response to DNA damage, p53 produces *lincRNA-p21*, *lincRNA-ROR*, and *PANDA* to regulate apoptosis. The image was self-created using BioRender.com.

**Table 1 ncrna-11-00009-t001:** Identified lincRNAs associated with *BRCA1* in breast cancer and their descriptive roles and expression patterns.

LincRNAs Associated with *BRCA1*	Roles	Expression Patterns	Ref.
** *NBR2* **	Regulates glucose metabolism, proliferation, and survival in breast cancer	Dysregulated in various cancers, including breast cancer	[74]
** *lincRNA-BC2* **	Upregulated in breast cancer and may influence oncogenic function	Significantly higher in breast cancer tissues	[32]
** *lincRNA-BC4* **	Linked to advanced breast cancer grade; potential tumor promoter	Downregulated in cancer tissues	[32]
** *lincRNA-BC5* **	Correlates with age and pathological stage; involved in tumorigenesis	Upregulated more than 2-fold in cancer samples	[32]
** *lincRNA-APOC1P1-3* **	Inhibits apoptosis and is associated with tumor size and hypomethylation	Overexpressed in breast cancer tissues	[75]
** *RP5-1198O20* **	Associated with breast cancer survival; potential biomarker	Significant expression changes noted	[75]
** *GATA3-AS1* **	Regulates *GATA3*, a key transcription factor implicated in tumor growth	Overexpressed in breast cancer	[75]
** *RP11-279F6* **	Interacts with transcription factors influencing cancer progression	Aberrantly expressed in breast cancer	[76]
** *Loc554202* **	Oncogenic role: promotes proliferation and inhibits apoptosis	High expression linked to aggressive tumor behavior	[77]
** *HOTAIR* **	Involved in chromatin remodeling and gene silencing; promotes metastasis	Upregulated in various breast cancer subtypes	[77]
** *MALAT1* **	Regulates alternative splicing and cell migration associated with metastasis	Overexpressed in breast cancer tissues	[77]
** *TUG1* **	Implicated in cell proliferation and migration; potential prognostic marker.	Expression levels vary across molecular breast cancer subtypes	[77]
** *LINC00152* **	Promotes cell proliferation and invasion; linked to poor prognosis	Frequently upregulated in breast cancer	[77]
** *NEAT1* **	Plays a role in nuclear body formation and stress response, linked to tumor aggressiveness	Increased expression observed in breast cancer	[77]
** *UCA1* **	Involved in cell cycle regulation and apoptosis inhibition	Overexpressed in breast cancer tissues	[77]
** *PVT1* **	Regulates myelocytomatosis (MYC) oncogene expression; involved in cell proliferation	Frequently upregulated in breast cancer	[77]
** *LINC00261* **	Modulates immune response and tumor microenvironment interactions	Expression altered significantly in tumors	[77]
** *LINC00673* **	Associated with EMT processes	Elevated levels found in aggressive breast cancer	[77]
** *LINC00467* **	Implicated in regulating apoptosis and cell cycle progression	Expression changes noted between normal and tumor tissues	[77]
** *LINC01554* **	Potentially involved in chemoresistance mechanisms	Expression varies significantly across different subtypes of breast cancer	[77]

## Data Availability

Not applicable.

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
