# Peer review of "Long Intergenic Non-Coding RNAs and BRCA1 in Breast Cancer Pathogenesis: Neighboring Companions or Nemeses?"

_ncrna, 2025, doi:10.3390/ncrna11010009_

Round 1
Reviewer 1 Report
Comments and Suggestions for Authors
The manuscript provides a comprehensive review of lincRNAs and their roles in BC, particularly focusing on their association with BRCA1 mutations. The paper highlights the potential of lincRNAs in regulating gene expression, influencing breast cancer progression, and serving as biomarkers or therapeutic targets, which holds translational value. The manuscript systematically addresses lincRNA classification, mechanisms, and their roles in breast cancer diagnosis, prognosis, and therapy. It is well-organized and extensively cites relevant literature. Here is my comments:
-
The abstract is overly detailed and lacks focus. Consider simplifying it to concisely present the background, key points, and contributions of the review.
-
The term "lincRNAs" is explained multiple times in the manuscript, which may cause redundancy. Define the term clearly on its first appearance and use the abbreviation consistently thereafter.
-
While the manuscript reviews the roles of lincRNAs in breast cancer, it lacks experimental or statistical data to support the discussed mechanisms and pathways. Although it is a review, adding illustrative figures or summary tables highlighting key findings from cited studies would enhance clarity and impact.
-
Some conclusions lack direct supporting references. For example, the discussion of specific lincRNAs in breast cancer metastasis would benefit from more detailed references to relevant studies.
-
Several sections, particularly those describing lincRNA classification and functions, include repetitive statements. Streamlining these sections would improve readability and maintain reader engagement.
-
The English language in the manuscript is generally clear and understandable
Author Response
We sincerely thank the reviewer for the constructive comments towards improving our manuscript.
- The abstract is overly detailed and lacks focus. Consider simplifying it to concisely present the background, key points, and contributions of the review.
Response: We have revised the abstract to improve its clarity and focus. The updated version succinctly presents the background and emphasizes the contributions of our review without unnecessary detail.
- The term "lincRNAs" is explained multiple times in the manuscript, which may cause redundancy. Define the term clearly on its first appearance and use the abbreviation consistently thereafter.
Response: We have addressed this concern by clearly defining "lincRNAs" at its first mention in the manuscript. Subsequent references to lincRNAs utilize the abbreviation consistently throughout the text, eliminating redundancy and enhancing readability.
- While the manuscript reviews the roles of lincRNAs in breast cancer, it lacks experimental or statistical data to support the discussed mechanisms and pathways. Although it is a review, adding illustrative figures or summary tables highlighting key findings from cited studies would enhance clarity and impact.
Response: We appreciate this suggestion and have incorporated several illustrative figures and a table that synthesize key findings from relevant studies on lincRNAs in breast cancer. These additions provide visual clarity and strengthen our discussion on mechanisms and pathways, enhancing our review's overall impact.
- Some conclusions lack direct supporting references. For example, the discussion of specific lincRNAs in breast cancer metastasis would benefit from more detailed references to relevant studies.
Response: We have carefully reviewed our conclusions and added appropriate citations to support our claims regarding specific lincRNAs involved in breast cancer metastasis.
- Several sections, particularly those describing lincRNA classification and functions, include repetitive statements. Streamlining these sections would improve readability and maintain reader engagement.
Response: We have revised sections that contained repetitive statements. By consolidating similar information and rephrasing where necessary, we have improved the flow of these sections.
- The review focuses heavily on summarizing existing findings without critically analyzing gaps in the field or offering detailed suggestions for future research directions. Including a dedicated section on unresolved questions and challenges would strengthen the paper's academic value.
Response: The potential of lincRNAs translational application and clinical utility is thoroughly discussed in sections 7 and 8, while current challenges is discussed in section 9 and future directions are discussed in section 10.
Reviewer 2 Report
Comments and Suggestions for Authors
The title is catchy, as is what was intended to be the subject of this review, which instead turns out to discuss lincRNAs from a more general point of view of breast cancer.
I would have appreciated a greater adherence to the title. In developing the manuscript, the authors lost sight of the title and purpose of the work.
It cannot be accepted in its present form.
The authors should reorganize the work completely reeling off those truly associated with BRCA1, or rework it based on a more generalized concept of lincRNAs in breast cancer. In this case, I think work could be considered.
Major revision
The manuscript focuses on lincRNAs in BC (paragraphs 5-16), but little space is left for those associated with BRCA1 (e.g. paragraph 17).
Minor revision
1. The abstract is too generic, and too introductory to the topic, without highlighting what should emerge from the review. Which lincRNAs are crucial and in association with what?
2. The data reported in the Introduction needs to be updated, as well as reference 1 (Global Cancer Statistics 2022). Figure 1 needs improvement, the X-axis does not show the full names, etc.
3. State what illustration software or tool the authors used to draw Figure 2 BioRender for example…?)
4. The sentence on line 160 is a repeated sentence.
5. The 5 paragraph is too generic. The following ones could be better organized as subparagraphs.
6. Punctuation mistakes.
7. English needs improvement.
Comments on the Quality of English LanguageEnglish needs improvement.
Author Response
We sincerely appreciate the reviewers' insightful comments and suggestions regarding improving our manuscript.
Major Revision
The manuscript focuses on lincRNAs in BC (paragraphs 5-16), but little space is left for those associated with BRCA1 (e.g., paragraph 17).
Response: We acknowledge the importance of discussing lincRNAs associated with BRCA1 in greater detail. We have expanded section 4 and Table 1 on BRCA1-related lincRNAs, integrating relevant findings and their implications in breast cancer (BCa).
Minor revision
- The abstract is too generic, and too introductory to the topic, without highlighting what should emerge from the review. Which lincRNAs are crucial and in association with what?
Response: The focus of the review is to discuss the broad landscape of lincRNAs in relation to BRCA-1 in BCa pathogenesis. While we acknowledge there is no defined specific lincRNAs within the pool, this subject fundamentally needs further experimental validated research in discovery, validation and deepen our insights into the precise roles played by the lincRNAs, particularly in relation to BRCA1 in BCa pathogenesis.
2. The data reported in the Introduction needs to be updated, as well as reference 1 (Global Cancer Statistics 2022). Figure 1 needs improvement; the X-axis does not show the full names, etc.
Response: We have updated the introduction with the most recent statistics and relevant literature. Reference 1 (Bray, F.; Laversanne, M.; Sung, H.; Ferlay, J.; Siegel, R.L.; Soerjomataram, I.; Jemal, A. Global cancer statistics 2022: GLOBOCAN estimates of incidence and mortality worldwide for 36 cancers in 185 countries. CA: A Cancer Journal for Clinicians 2024, 74, 229-263, doi:https://doi.org/10.3322/caac.21834.) has been revised accordingly. Additionally, we have improved Figure 1 by ensuring all axis labels are fully spelt out for clarity.
3. State what illustration software or tool the authors used to draw Figure 2 (BioRender, for example).
Response: We have added a note in the figure legend for Figure 2 specifying that BioRender was used for its creation.
4. The sentence on line 160 is a repeated sentence.
Response: We have corrected this issue by removing the repeated sentence and ensuring that the text flows smoothly without redundancy.
5. The fifth paragraph is too generic. The following ones could be better organized as subparagraphs.
Response: This has been restructured and subsequent sections have been converted into subparagraphs, 5.1 and 5.2 for better clarity and coherence.
6. Punctuation mistakes.
Response: We have thoroughly proofread the manuscript to correct all punctuation errors identified by the reviewer.
7. English needs improvement.
Response: We have revised the manuscript for grammatical accuracy and clarity, and if necessary, English can be further improved.
Round 2
Reviewer 1 Report
Comments and Suggestions for Authors
NA
Comments on the Quality of English LanguageNA
Author Response
We appreciate your thoughtful feedback and the opportunity to revise our manuscript. We have consulted a professional language editor to enhance the clarity and quality of our work. This process has significantly improved the manuscript, addressing the concerns you raised and ensuring it meets the high standards of your esteemed journal.
We believe our revised manuscript is stronger and more deserving of publication in your reputable journal.
Thank you for your time and consideration.
Reviewer 2 Report
Comments and Suggestions for Authors
The authors have made a significant effort to reorganize the work, which now appears more focused. I appreciate this new version.
However, the format is full of errors. It shows a lack of care in assembling the manuscript: the legends, the figures, the punctuation, and the references, not to mention the randomly distributed abbreviations.
The lack of care shown does not yet allow us to accept the manuscript, which appears hastily reassembled.
I suggest the authors write the next version with more care, attention, and consideration towards this journal. If this is not done, the manuscript will not be further revised.
Minor revision
Distribute the manuscript evenly across the margins. There are many punctuation errors. The manuscript has not been carefully revised. Also, English needs improvement. Genes must ALWAYS be indicated in italics.
The sequence of references needs to be reviewed and corrected.
Some examples:
lines 34, 40, 43, and so on…136, 140, 141: punctuation errors, lack of space e.g. (Figure 1)[1], and many more.
line 74: HER2-, the minus goes to the apex, and luminal A/B- ….without -. Please recheck the text and correct other similar errors.
line 102: the legend of Figure 2 does not have the correct format required by the journal. It is hastily done, without proper punctuation. In addition, the abbreviations are poorly described and confusing.
Line 117: Abbreviations should be indicated the first time that term is used, such as PR, ER, and TNBC, and always used thereafter. Also, lines 112 and 158….and other similar errors….lines 173 (title),179…205..217..262, 341, 346, 354… to report some.
Line 119: English errors, the manuscript must be checked.
Line 144: BRCA1-mutated
Line 309: the "LincRNAs as regulators of BRCA1" paragraph could be included as 5.3
Line 361: paragraph 6. Please, correct others accordingly.
Comments on the Quality of English LanguageEnglish needs improvement.
Author Response
In preparing this revised manuscript, we consulted a professional language editor to ensure the highest quality of language and clarity, significantly enhancing the manuscript's readability and overall presentation.
We appreciate the reviewer's feedback and suggestions, which have been invaluable in refining our manuscript. Below are detailed responses to each of the comments:
Comments and Suggestions for Authors
- Comment: The authors have made a significant effort to reorganize the work, which now appears more focused. I appreciate this new version.
Response: We appreciate the reviewer's feedback and suggestions, which have been invaluable in refining our manuscript.
- Comment: However, the format is full of errors. It shows a lack of care in assembling the manuscript: the legends, the figures, the punctuation, and the references, not to mention the randomly distributed abbreviations.
Response: We acknowledge the previous errors and have thoroughly reviewed and corrected the manuscript. We have ensured that all legends and figures are appropriately formatted, punctuation is accurate, references are correctly sequenced, and abbreviations are consistently used and defined.
- Comment: The lack of care shown does not yet allow us to accept the manuscript, which appears hastily reassembled. This is annoying.
Response: We understand the concern and have carefully prepared this revised version. We have meticulously reviewed every aspect of the manuscript to ensure accuracy, consistency, and adherence to the journal's guidelines.
- Comment: I suggest the authors write the next version with more care, attention, and consideration towards this journal. If this is not done, the manuscript will not be further revised. The authors show too much haste and little respect for the review process.
Response: We appreciate this feedback and have implemented it entirely. This revised version reflects our commitment to quality and respect for the review process.
Minor revision
- Comment: Distribute the manuscript evenly across the margins. There are many punctuation errors. The manuscript has not been carefully revised. Also, English needs improvement. Genes must ALWAYS be indicated in italics.
Response: We have adjusted the margins to ensure even distribution and corrected all punctuation errors to improve readability. Also, we have refined the English throughout the manuscript and ensured that all gene names are correctly formatted in italics.
- Comment: The sequence of references needs to be reviewed and corrected.
Some examples:
lines 34, 40, 43, and so on…136, 140, 141: punctuation errors, lack of space e.g. (Figure 1)[1], and many more.
Response: We have thoroughly reviewed and corrected the punctuation errors and reference sequence to ensure accuracy and consistency.
- Comment: line 74: HER2-, the minus goes to the apex, and luminal A/B- ….without -. Please recheck the text and correct other similar errors.
Response: We have corrected these terms as suggested, ensuring accuracy in notation.
- Comment: line 102 the legend of Figure 2 does not have the correct format required by the journal. It is hastily done, without proper punctuation. In addition, the abbreviations are poorly described and confusing.
Response: We have revised the legend of Figure 2 to meet the journal's requirements, ensuring proper punctuation and clear descriptions of abbreviations.
- Comment: Line 117: Abbreviations should be indicated the first time that term is used, such as PR, ER, and TNBC, and always used thereafter. Also, lines 112 and 158….and other similar errors….lines 173 (title),179…205..217..262, 341, 346, 354… to report some.
Response: We have addressed all mentioned corrections, ensuring that the manuscript is thoroughly revised and polished
- Comment: Line 119 English errors, the manuscript must be checked.
Response: We have refined the English errors throughout the manuscript
- Comment: Line 144 BRCA1-mutated
Response: The term "BRCA-mutated” has been used in recent literature, such as by Arun et al. in their discussion of BRCA-mutated breast cancer, highlighting its established usage in the field. However, we are open to adjusting the terminology if you find it incoherent or unclear in the context of our article.
(BRCA-mutated breast cancer: the unmet need, challenges and therapeutic benefits of genetic testing. Br J Cancer 131, 1400–1414 (2024). https://doi.org/10.1038/s41416-024-02827-z).
- Comment: Line 309 the "LincRNAs as regulators of BRCA1" paragraph could be included as 5.3
Response: The mentioned paragraph has been included as 5.3
- Comment: Line 361: paragraph 6. Please, correct others accordingly.
Response: Paragraph 6 has been thoroughly refined.
We thank you for your feedback. We are confident that this revised version meets the high standards of your journal.
